# Satellite-Based Localization of IoT Devices Using Joint Doppler and Angle-of-Arrival Estimation

Iza S. Mohamad Hashim * and Akram Al-Hourani

School of Engineering, City Campus, RMIT University, Melbourne, VIC 3000, Australia;
akram.hourani.rmit@edu.au
* Correspondence: izahashim@ieee.org

**Abstract:** While global navigation satellite system (GNSS) technologies have always been the go-to solution for localization problems, they may not be the best choice for some Internet-of-Things (IoT) applications due to the incurred power consumption and cost. In this paper, we present an alternative satellite-based localization method exploiting the signature of Doppler shifts and angle-of-arrival measurements as seen by a low-Earth-orbit (LEO) satellite. We first derive the joint likelihood function of the measurements, which is represented as a combination of three Gaussian distributions. Then, we show that the maximum likelihood problem reduces to a more-efficient mean squared error minimization in the Gaussian case as inferred from real measurements we collected from low-Earth-orbit satellite using a tracking ground station. Thus, we propose utilizing a stochastic optimizer to search for the global minimum of the mean squared error, which represents the location of the ground IoT device as estimated by the satellite platform. The emulated results show that the IoT device localization, in such a realistic model, can be performed with sufficient accuracy for IoT applications.

**Keywords:** localization; satellite; Internet of Things; AoA; Doppler; likelihood; low Earth orbit

## 1. Introduction

Internet-of-Things (IoT) technologies have revolutionized industry operations across the globe. According to the 3GPP standardization body, IoT deployments can be categorized into two groups: (i) terrestrial IoT, which is served by typical terrestrial cellular stations, and (ii) non-terrestrial network (NTN) IoT, which cover unmanned aerial vehicles (UAVs) and satellite networks. Due to the extensive use of IoT devices, location awareness is deemed to be beneficial in many IoT applications, with examples in asset management, tracking, logistics, environmental monitoring, and automated agriculture [1,2]. Parallel to the growing IoT demand, the rapid advances in the space industry have increased the feasibility of IoT-over-satellite connectivity [3], thus extending the network coverage for new regions that were difficult to reach using typical terrestrial technologies. Whenever a localization solution is needed, the common strategy is to utilize the Global Navigation Satellite System (GNSS) services such as GPS, Galileo, and BeiDou, while the GNSS possesses very precise positioning capabilities, it is not always the best solution if IoT hardware constraints are a limiting factor. Such constraints are imposed by limited energy storage, computational resources, GNSS module cost, and communication overhead [1,4]. Accordingly, alternative localization methods tailored for IoT applications are intriguing to investigate, especially if they can be utilized alongside existing communication methods without the need for a dedicated hardware module. They are particularly useful for any legacy IoT devices that are not equipped with a GNSS chip.

Doppler shifts in the carrier signal have long been used in satellite-based geolocalization, where—due to the vast velocity of satellite platforms—Doppler has a distinct value dominated by the deterministic satellite movement, which is much larger than the unknown variations caused by IoT device mobility. For example, the ARGOS satellite

system is one of the first prominent applications of Doppler measurements for locating ground transmitters [5,6]. Other than ARGOS, Doppler geolocation has also been used in the COSPAS/SARSAT and TIROS-N [7] polar orbiting satellites, which primarily operate as search-and-rescue mission satellites [8].

In addition to Doppler measurements, angle of arrival (AoA) was one of first methods used in finding radio emitters, originally was performed by mechanically steering the receiving antenna. With the advancement in digitally controlled phase shifters and in fully-digital radio heads, multi-antenna AoA estimation can now achieve very accurate results [9–11] especially with the use of super-resolution methods such as MUSIC [12], ESPRIT [13], and weighted subspace fitting [14,15]. AoA does not require clock synchronization with the transmitter, thus making it simpler to implement [10] than the time-of-arrival (ToA) method used in the GNSS. Rather than using AoA and Doppler measurements individually, it is compelling to investigate the effect of fusing both AoA and Doppler measurements to obtain higher accuracy. However, merging both of the measurements in the estimation problem is challenging [16]. Several algorithms are used to tackle the joint estimation problem, such as the maximum likelihood estimator (MLE) [16–18] and pseudolinear estimator, particularly the weighted instrumental variable estimator (WIVE) [19].

In this study, we investigated the performance of multi-antenna satellite-based localization by combining (i) the Doppler shift frequency signature with (ii) AoA measurements obtained at a low-Earth-orbit (LEO) satellite while it passes over an IoT device. In order to model realistic Doppler error behavior, we collected real transmissions from a LEO orbiting satellite, namely from the National Oceanic and Atmospheric Administration (NOAA) satellite, NOAA-15, using a ground station with an automated motorized tracking antenna. For precise Doppler error quantification, we utilized a software-defined radio locked to a GPS-disciplined oscillator. Based on this developed model, we emulated extensive satellite passes and used a joint Doppler shift and AoA measurement likelihood function to estimate the ground IoT device location. We applied stochastic optimization to find the global maximum of the likelihood function, which corresponds to the best location estimate of the ground IoT device. The contributions of the study are outlined as follows:

- A satellite-based localization method using joint Doppler and AoA measurements received from the ground IoT device is proposed.
- The likelihood function of Doppler and AoA measurements based is derived on the Gaussian error and estimated Kent error distributions, respectively.
- The Doppler measurement error was investigated using real measurements from LEO satellites.
- The localization performance behavior against varying Doppler and AoA error deviations is illustrated.

## 2. Related Works

IoT over NTN is gradually becoming a viable solution due to market newcomers and cheaper access to space. This is gradually covering the gap left by terrestrial networks, due to both technical and financial limitations. Figure 1 illustrates the overview of the NTN category, which includes (i) airborne platforms that are deployed in the troposphere, such as unmanned aerial vehicles (UAVs), or stratosphere, like high-altitude platforms (HAPs), and (ii) spaceborne platforms such as satellites at various altitudes. The satellite altitude ranges from low Earth orbits (180–2000 km) to geosynchronous equatorial orbits (GEOs) at around 35,786 km [20]. A spaceborne LEO NTN is able to provide a wider coverage with a beam footprint of approximately 100–1000 km in diameter compared to that of the airborne NTN with a diameter of only around 5–50 km [21]. However, the spaceborne link is more susceptible to losses, such as the atmospheric effect, and free space path loss due to its high altitude.

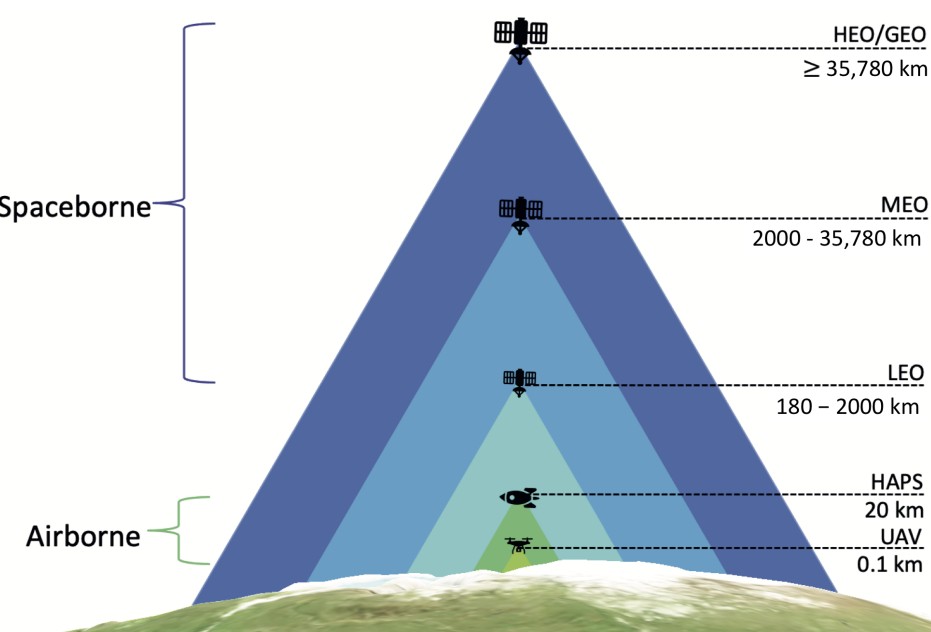

**Figure 1.** Overview of the non-terrestrial network (NTN): spaceborne and airborne.

Having location information is advantageous, or even vital, in some IoT applications such as mining, agriculture, and environmental monitoring. The GNSS is often the established solution for localization problems given its ubiquity and high accuracy. However, for some IoT use cases, especially in low-cost and energy-limited IoT devices, the GNSS could be substituted by direct localization performed by the satellite itself on the uplink communication signal from the IoT device.

One of the suitable satellite-based localization methods is the use of Doppler measurements [22], which only works for non-geostationary satellites. Two available algorithms can be used in previously mentioned ARGOS systems: least-squares and Kalman filtering [5]. The least-squares method has not been changed since 2007 [23], in this scheme the localization algorithm needs at least four uplink transmissions of the ARGOS message that contains several Doppler measurements during the initialization procedure [5]. After the initialization, it requires at least two uplink transmissions per satellite pass to find the intersection of the Doppler solution, which provides both the actual and mirror positions. The Kalman filter algorithm also requires four transmissions during initialization, but it only needs a single uplink transmission per satellite pass, where it uses the interactive multiple models (IMMs) of the unscented Kalman filters (UKFs) to handle multiple hypotheses when combining measurements with multiple behaviors [6].

Other than Doppler measurements, angle-of-arrival (AoA) measurements can also be used to localize ground devices using satellites. One of the common methods for localization using AoA measurements is triangulation. Typically, the angle estimation is approximated with an assumption of a flat Earth surface for a low altitude observer such as an unmanned aerial vehicle (UAV), since the altitude is insignificant compared to the Earth's radius. For a low-Earth-orbit (LEO) satellite observer, the work in [10] utilizes spherical trigonometry and singular value decomposition (SVD) to localize a ground target when more than two AoA measurement points are available. The analysis shows that the higher the number of AoA measurement points, the lower the root mean squared localization error. Furthermore, it is crucial to consider the geolocation bias in the target position estimate due to high non-linearity in the AoA measurements [24]. To account for the bias error, several compensation methods have been presented in [11,25,26]. For example, a reduced-bias pseudolinear estimator in [26] can decrease the bias by acquiring sufficient measurements where the bias is dependent on the correlation vector of the measurement matrix. Moreover, both works in [11,25] apply a Taylor-series expansion to estimate the

bias; where [11] applies it directly to the target position estimate, Ref. [25] applies it to the maximum likelihood (ML) cost function up to the second order.

To further improve satellite-based localization performance, multiple types of measurements could be fused to compensate for the shortcomings of each measurement type. However, combining different types of measurements poses some degree of difficulty. For example, the work in [16] uses a maximum likelihood estimator (MLE), which is based on importance sampling to estimate the Doppler and AoA measurements. MLE is chosen to achieve higher accuracy compared to the other methods, such as using two sequential one dimension (1D) optimization problems [27] or the use of a combination of two 1D frequency-multiple signal classification/estimation of signal parameters via rotational invariance techniques (MUSIC/ESPRIT) and one 1D space-MUSIC/ESPRIT to form a tree-structured frequency–space–frequency (FSF) MUSIC/ESPRIT algorithm [28]. For the first method, the 1D sequential optimization is run one-at-a-time, while the other parameters are fixed. Then, the process is repeated to optimize the previously fixed parameters and set the other parameters as fixed variables. However, this method is an iterative process and only converges to a local maximum [27].

Furthermore, the latter approach [28] can estimate the Doppler and angle of two sources. If the Doppler or angle of the two sources is very similar, the estimation is only possible assuming they are using the same radio resources (in this case, the same frequency band) and that the other parameters are well separated from the closed one. Furthermore, it is essential to know or estimate a priori the number of sources with the same angle or Doppler. In [16], a maximum joint likelihood of Doppler and AoA is proposed. Formulating a closed-form solution of the joint likelihood function does not seem to be possible [16]. Therefore, the work in [16] suggests using a global maximization method for the nonlinear likelihood optimization. In this work, [16], the Doppler and AoA are estimated from the received complex signal. Other than that, WIVE can also be used to eliminate the correlation between the measurement matrix and the pseudolinear noise vector by exploiting the instrumental variables. In [19], an improved version of the weighted instrumental variable estimator (I-WIVE) is proposed to cater to large measurement noise as the WIVE algorithm is only asymptotically unbiased when small measurement noise is present. I-WIVE is developed by estimating the bias of WIVE and subtracting it from the WIVE estimate. It is proven from the simulation results that the I-WIVE has better performance than the WIVE.

Most of the work presented in the literature for the joint Doppler and AoA measurements revolves around the estimation of the measurements [16,27,28] instead of localizing the source directly. The satellite-based localization literature [5,6,10] typically reports the usage of either the Doppler shift frequency or AoA measurements. Therefore, to address the gap, we present a satellite-based ground IoT device localization framework using both the Doppler and AoA measurements by expressing a likelihood function that characterizes the joint measurements given the proposed ground IoT device's location.

## 3. System Model

In order to generate a Doppler signature, the satellite(s) need to be in relative motion to the user. As such, we focus on low-Earth-orbit (LEO) satellite constellations, assuming that the satellites have accurate knowledge of their own position/orbit. In this scenario, the satellites act as anchor nodes that attempt to localize the ground IoT devices by utilizing both the angle-of-arrival (AoA) and Doppler measurements. This section presents the constellation geometry, Doppler measurement model, and angle-of-arrival measurement model.

### 3.1. Constellation Geometric Model

Although the proposed framework is not restricted to a particular geometric setup, we consider the common circular polar orbit because of its near-uniform distance to the Earth's surface throughout its orbital period. For the constellation model, we adopt the popular Walker-star pattern which is widely used in telecommunication applications. The Walker-star pattern has three key parameters [29]: (i) the total number of satellites $N$, (ii)

the number of orbital planes $P$, and (iii) the phasing parameter $F$. The phasing parameter is defined as an integer within $0 \leq F \leq (P-1)$ representing the relative offset between satellites in two subsequent orbital planes. In the Walker-star constellation, the orbital planes are uniformly rotated around the Earth's axis; thus, the right ascension of the ascending nodes (RAAN) for an orbital plane $j \in [1, P]$ is given by [30],

$$\Omega_j = \frac{\pi}{P}(j-1),\tag{1}$$

and the initial true anomaly $\nu$ is given by [31],

$$\nu_{jl} = \frac{2\pi}{S}(l-1) + \frac{2\pi F}{N}(j-1),\tag{2}$$

where $l$ is the order within the orbital plane, i.e., $l \in (l, S)$, and $S = N/P$ is the total number of satellites on the orbital plane. An example of the satellite orbits following the Walker-star pattern is shown in Figure 2.

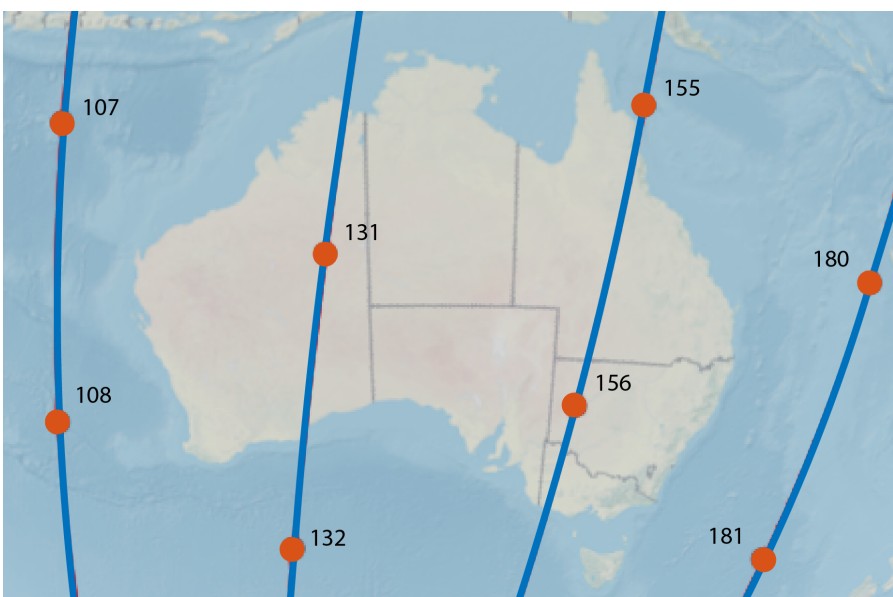

**Figure 2.** Example of satellite constellation orbits using Walker-star pattern with 12 orbital planes and 24 satellites distributed equally on a single plane.

### 3.2. Doppler Model

As we aim to make use of the Doppler shift pattern in the localization framework, it is important to adopt a practical measurement model which can also be empirically validated. Without loss of generality, we consider a stationary (or slowly moving) target where its ground velocity is negligible in comparison with the motion of the satellite with respect to the Earth's fixed inertial frame (ECI). The classical Doppler shift can be found from the radial velocity, which is computed by taking the derivative of the slant distance as follows,

$$v_{\mathrm{r}} = \frac{\rho(t + \Delta t) - \rho(t)}{\Delta t},\tag{3}$$

where $\rho$ is the slant distance between the satellite and the ground IoT device, $\Delta t$ is a small time step, and $t$ is the time variable. Accordingly, the true mean classical Doppler shift is obtained as $\mu_{\mathrm{d}} = -\frac{v_{\mathrm{r}}}{c}f$, where $c$ is the propagation speed (light speed in free-space), $f$ is the center operating frequency, and $v_{\mathrm{r}}$ is the true radial velocity as obtained in (3).

In addition, we also investigated the effect of modeling the Doppler measurements using the relativistic Doppler effect for a receding source as follows

$$f_r = \frac{f}{\gamma(1 + v_r/c)} - f, \tag{4}$$

where $\gamma = 1/\sqrt{(1 - v^2/c^2)}$ is the Lorentz factor, and $v$ is the velocity, which can be obtained by differentiating the position vector considering the satellite as the inertial frame of reference. The time dilation term is the difference between the classical and the relativistic Doppler.

From the plot in Figure 3, we can observe that there is a difference between the two calculations with a 0.62 Hz maximum relative error. Accordingly, in this study, we maintained the usage of the classical Doppler formula for the Doppler measurements.

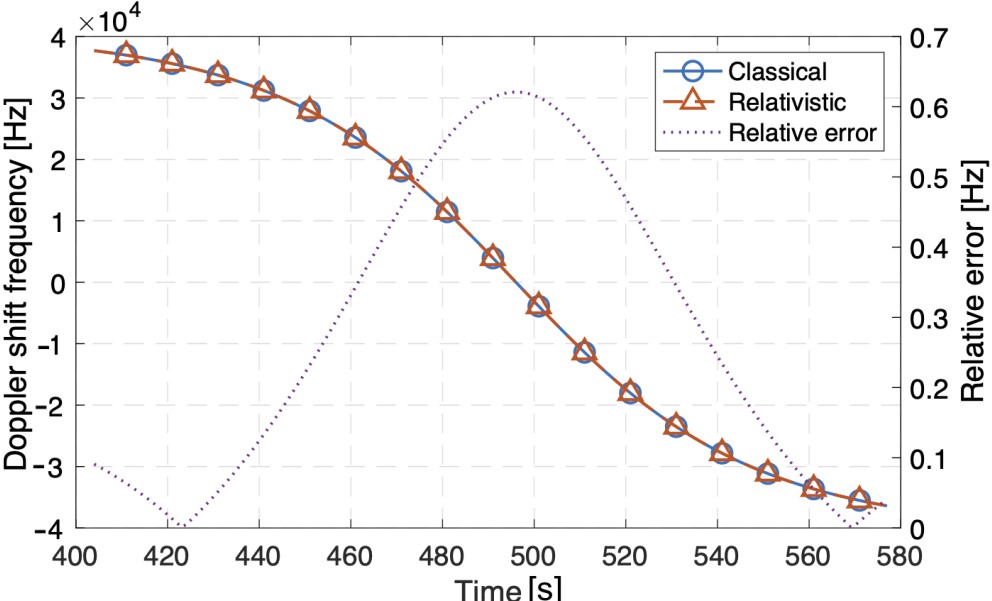

**Figure 3.** Comparison between the classical and relativistic Doppler shift frequency measured from a polar orbiting LEO satellite at an altitude of 833 km. The highest relative error is at the inflection point (around 0.62 Hz).

We demonstrate in Figure 4 the Doppler patterns for different LEO orbit altitudes. It is evident that the Doppler shift has a distinct pattern at different altitudes. As the satellite approaches the ground transmitter, the Doppler shift curve slope increases, and it is 0 when the distance between the satellite and the ground IoT device is the smallest, as the transverse velocity is null. In contrast, as the satellite recedes, the Doppler shift curve slope decreases. The instance point where the slope changes is called the *inflection point*, which corresponds to the point where the satellite is nearest to the ground IoT device.

For the measurement noise model of the Doppler, we adopted the Gaussian distribution based on its wide use in the literature [32–34] and also based on our own direct measurements from the NOAA satellites. In this model, the Doppler error can be formulated as a zero-mean Gaussian distribution with a certain standard deviation, $\sigma_d$ as shown in the upcoming Section 6.1. The error due to atmospheric effects can be excluded from the model due to its negligible effect compared to the satellite's velocity [34,35]. Thus, the Doppler measurements with the presence of error can be modeled as,

$$f_d \sim \mathcal{N}\left(\mu_d, \sigma_d^2\right). \tag{5}$$

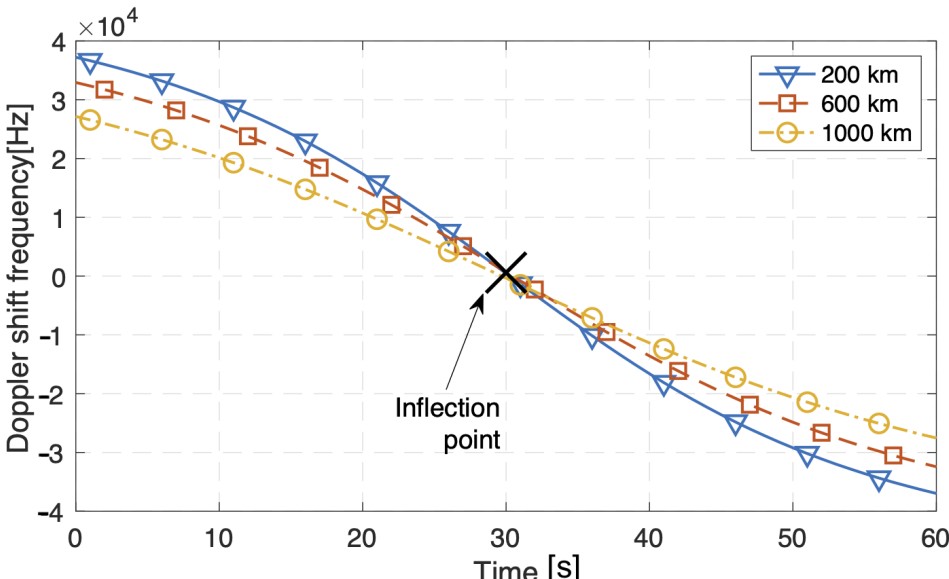

**Figure 4.** Doppler shift frequency at different LEO polar orbit altitudes (ranges from 200 to 1000 km).

### 3.3. Angle-of-Arrival Model

A generalized direction-finding array should be capable of estimating the ground emitter direction (IoT device location) in a 2D (dimensional) angular space. We consider the azimuth, $\Phi$, and the off-nadir angle, $\Theta$, centered at the satellite in the north–east–down (NED) frame, where the xNorth axis is parallel to the Earth's surface pointing north along the meridian of longitude, the yEast axis is also parallel to the Earth's surface pointing east along the latitude, and the zDown axis is antiparallel to the Earth's surface normal at the satellite subpoint. The NED frame is shown in Figure 5, where $\Phi$ is the angle obtained from the xNorth positive axis to the projection of the signal source on the xNorth–yEast plane, and off-nadir angle $\Theta$ is the angle computed from the zDown axis (i.e., pointing towards the nadir) towards the signal's source.

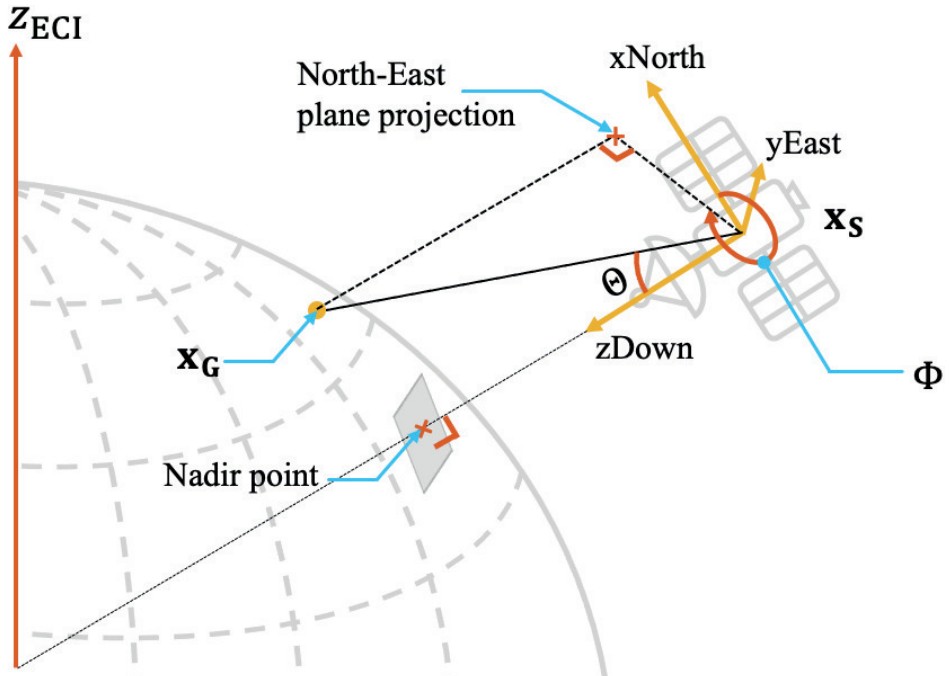

**Figure 5.** Illustration of the ground IoT device's angle of arrival measured at satellite in north–east–down (NED) frame. Antenna boresight is assumed to be oriented towards the nadir.

To convert the Earth-centered Earth-fixed (ECEF) coordinates to the NED coordinates so that the origin is defined at the satellite instead of the Earth's center, we need to apply a coordinate transformation matrix, $\mathbf{M}$ [36] to the position vector, $\mathbf{x_a} = [x_a \ y_a \ z_a]$, where $\mathbf{M}$ is defined as follows:

$$\mathbf{M} = \begin{bmatrix} -\sin\phi_S\cos\lambda_S & -\sin\phi_S\sin\lambda_S & \cos\phi_S \\ -\sin\lambda_S & \cos\lambda_S & 0 \\ -\cos\phi_S\cos\lambda_S & -\cos\phi_S\sin\lambda_S & -\sin\phi_S \end{bmatrix}, \tag{6}$$

where $\phi_S$ is the latitude and $\lambda_S$ is longitude of the satellite and position vector is calculated as $\mathbf{x_a} = \mathbf{M}(\mathbf{x_G} - \mathbf{x_S})$. $\mathbf{x_S}$ and $\mathbf{x_G}$ are the satellite and ground IoT device coordinates vectors in Earth-fixed Earth-centered frames, respectively.

For the AoA measurement error, we adopt the common Kent distribution model which (for small errors) can be approximated into Gaussian additive measurement noise for both azimuth and elevation angles as follows [37],

$$\mathcal{K}(\Phi, \Theta; \kappa, \beta, I_3) \approx$$
$$\mathcal{N}\left(\Phi; 0, \frac{1}{\kappa - 2\beta}\right)\mathcal{N}\left(\Theta; 0, \frac{1}{\kappa + 2\beta}\right), \tag{7}$$

where $\kappa$ is the concentration parameter, $\beta$ is the ovalness parameter such that $0 \leq \beta < \kappa/2$, and $I_3$ is the identity matrix of size 3. In the context of the AoA measurement, $\kappa$ relates to the spread of the measurements, whereas $\beta$ represents the ellipticity of the measurement distribution. Alternatively, we can directly define $\sigma_\Phi, \sigma_\Theta$ as the standard deviation of azimuth and off-nadir measurement errors in the normal distribution, where $\sigma_\Phi = \frac{1}{\kappa - 2\beta}$ and $\sigma_\Theta = \frac{1}{\kappa + 2\beta}$. Note that in this model, the AoA measurement error due to satellite attitude, refraction, and other errors are not considered. This is to isolate the impact of the measurement uncertainty only.

## 4. Likelihood Derivation

In this localization problem, we aim to estimate the *state* vector $\mathbf{x} = [\text{lat, lon}]^T$ which represents the unknown position (latitude and longitude) of a ground IoT device in the geodetic coordinates system. To do so, we utilize a series of three measurements, namely Doppler and angle of arrival (AoA) in 2D (azimuth and off-nadir), $\mathbf{Z}^{(\mathbf{k})} = [f_d^{(k)}, \Phi^{(k)}, \Theta^{(k)}]^T$, where $k$ is the discrete-measurement index.

### 4.1. Doppler and Angle-of-Arrival Likelihood Derivation

The Doppler likelihood of a single measurement $k$ is derived from the probability density function of the Gaussian distribution with a standard deviation of $\sigma_d$ as follows,

$$g_1(f_d^{(k)}|\mathbf{x}) = \mathcal{N}(f_d^{(k)}; \mu_d^{(k)}(\mathbf{x}), \sigma_d^2), \tag{8}$$

where $\mu_d^{(k)}(.)$ is the true Doppler. Next, we obtain the AoA likelihood function from the probability distribution function (PDF) of the Gaussian estimated Kent distribution assuming that the errors in the azimuth and off-nadir are independent,

$$g_2(\Phi^{(k)}, \Theta^{(k)}|\mathbf{x}) = \mathcal{N}\left(\Phi^{(k)}; \mu_\Phi^{(k)}(\mathbf{x}), \frac{1}{\kappa - 2\beta}\right)$$
$$\mathcal{N}\left(\Theta^{(k)}; \mu_\Theta^{(k)}(\mathbf{x}), \frac{1}{\kappa + 2\beta}\right), \tag{9}$$

where $\mu_\Phi^{(k)}$ and $\mu_\Theta^{(k)}$ are the true azimuth and the true off-nadir angle of arrival, respectively.

### 4.2. Joint Likelihood of Doppler and Angle of Arrival

Since the three measured variables are assumed as mutually independent, the joint likelihood of the Doppler and AoA is thus the product of the two likelihood functions, $p(\mathbf{Z}^{(k)}|\mathbf{x}) = g_1(f_{\mathrm{d}}^{(k)}|\mathbf{x}) \times g_2(\Phi^{(k)}, \Theta^{(k)}|\mathbf{x})$. Moreover, to incorporate measurements from multiple satellites, the likelihood of the $K$ measurements is computed as the product of a single measurement,

$$p(\mathbf{Z}^{(1:K)}|\mathbf{x}) = \prod_{(k)=1}^{(K)} p(\mathbf{Z}^{(k)}|\mathbf{x}), \tag{10}$$

where $\mathbf{Z}^{(1:K)} \equiv \{\mathbf{Z}^1, \mathbf{Z}^2, ..., \mathbf{Z}^{(k)}\}$, and $K$ is the total number of measurements taken from multiple satellites. Note that this computation method holds true under the assumption of having independent measurement errors for a number of observed measurements [38].

### 4.3. Minimizing Negative Log Likelihood

The likelihood function of the Gaussian distribution is represented as the Gaussian probability density function (PDF). In the following PDF, we express the exponential term as $h$ to simplify the expression as follows,

$$g(z) = \frac{1}{\sigma\sqrt{2\pi}} \exp(h), \text{where}, \tag{11}$$

$$h = -\frac{(z - \mu)^2}{2\sigma^2}.$$

From the likelihood function, we then reduce the problem to a mean squared minimization. Firstly, we take the natural logarithm of the likelihood function. Since both likelihood functions are Gaussian, the joint likelihood function can be expressed as follows,

$$
\begin{aligned}
-\ln\left(p\left(\mathbf{Z}^{(1:K)}|\mathbf{x}\right)\right) &= -\ln\left(\prod_{(k)=1}^{(K)}\left(g_1^{(k)} g_2^{(k)}\right)\right) \\
&\stackrel{(a)}{=} -\ln C \prod_{(k)=1}^{(K)} \exp\left(h_1^{(k)}\right) \prod_{(k)=1}^{(K)} \exp\left(h_2^{(k)}\right) \\
&\stackrel{(b)}{=} \sum_{(k)=1}^{(K)} h_1^{(k)} + \sum_{(k)=1}^{(K)} h_2^{(k)} + C \\
&\stackrel{(c)}{=} \sum_{(k)=1}^{(K)} \frac{\left(f_{\mathrm{d}}^{(k)} - \mu_{\mathrm{d}}^{(k)}(\mathbf{x})\right)^2}{2\sigma_{\mathrm{d}}^2} + \sum_{(k)=1}^{(K)} \frac{\left(\Phi^{(k)} - \mu_{\Phi}^{(k)}(\mathbf{x})\right)^2}{2\sigma_{\Phi}^2} + \\
&\quad \sum_{(k)=1}^{(K)} \frac{\left(\Theta^{(k)} - \mu_{\Theta}^{(k)}(\mathbf{x})\right)^2}{2\sigma_{\Theta}^2} + C,
\end{aligned}
\tag{12}
$$

where $f_{\mathrm{d}}$ is the Doppler measurement, $\Phi$ is the azimuth, $\Theta$ is the off-nadir angles of the AoA measurement, and $C$ is a constant term. Since we take the natural logarithm of the joint likelihood, the exponential in (a) can be canceled out, and its product can then be expressed as addition, as shown in (b). Note that the constant term of the PDF, $\frac{1}{\sigma\sqrt{2\pi}}$, is brought out from the expression, which will then be represented as $C$ in the following steps. Then, we substitute the exponential term from the Gaussian PDF in (11) to (b). Finally, the equation can be simplified as presented in (c). The goal is to maximize the likelihood function, which is equivalent to finding the minimum mean squared error of the negative log likelihood.

## 5. Localization Framework

In this framework, we aim to estimate the location of the ground IoT device **x** from the combination of Doppler and AoA measurements $\mathbf{Z}^{(1:K)}$ using the joint likelihood function. We utilize the stochastic optimization method to estimate the location.

In our problem, we aim to maximize the cost function that is represented by the likelihood function:

$$\tilde{\mathbf{x}} = \underset{\mathbf{x}}{\mathrm{argmax}}\left[ p\left( \mathbf{Z}^{(1:K)} | \mathbf{x} \right) \right], \tag{13}$$

where $\tilde{\mathbf{x}}$ represents the estimated state. However, since the likelihood functions are Gaussian, maximizing the Gaussian likelihood functions is equivalent to minimizing the mean squared error, as discussed in Section 4.3. The optimization of the mean squared error can then be expressed as follows,

$$\tilde{\mathbf{x}} = \underset{\mathbf{x}}{\mathrm{argmin}}\left[ \sum_{(k)=1}^{(K)} \frac{\left( f_{\mathrm{d}}^{(k)} - \mu_{\mathrm{d}}^{(k)}(\mathbf{x}) \right)^2}{\sigma_{\mathrm{d}}^2} + \right.$$
$$\sum_{(k)=1}^{(K)} \frac{\left( \Phi^{(k)} - \mu_{\Phi}^{(k)}(\mathbf{x}) \right)^2}{\sigma_{\Phi}^2} + \tag{14}$$
$$\left. \sum_{(k)=1}^{(K)} \frac{\left( \Theta^{(k)} - \mu_{\Theta}^{(k)}(\mathbf{x}) \right)^2}{\sigma_{\Theta}^2} \right].$$

The *C* constant is ignored in the minimizing computation as the constant does not change the point where the minimum error lies.

## 6. Experiment and Results

In the first part of the experiment, we collected approximately 12 min of Doppler measurements from the NOAA-15 satellite at five different passes to analyze the Doppler error measurements. From the analysis, we were able to gain an overview of the Doppler error measurements, thus helping us to accurately model the Doppler measurements. We then emulated both the Doppler and AoA measurements based on the previously discussed models. The simulated measurements were then used to localize the ground IoT device. We present the simulation results of the localization performance under multiple conditions in the upcoming subsection.

### 6.1. Doppler Error Measurements

To account for realistic Doppler measurements in our emulation, we investigated the Doppler error measurement distribution based on an actual received signal from a LEO satellite. Namely, we obtained the Doppler measurements from NOAA-15. The satellite follows a circular sun-synchronous near-polar orbit with an inclination of 98.7° and an altitude of 833 km. The satellite completes its orbit 14 times per day. It has an automatic picture transmission (APT) module operating at a very high frequency (VHF) of 137.62 MHz. Furthermore, it has an S-band transmitter that operates at 1702.5 MHz with an omnidirectional antenna. We selected NOAA-15 as it is one of the satellites used in the ARGOS system.

The setup diagram of the experiment is shown in Figure 6. We utilized the digital control interface to command the elevation–azimuth controller, which is connected to the antenna rotator, in order to keep track of the passing LEO satellite. The physical connection between the interface and the controller can be seen in Figure 7. There are two motors moving the VHF band antenna to move the antenna in both the azimuth and elevation planes. The top motor is for elevation, and the bottom is for azimuth, as observed in

Figure 8. The tracking uses the SGP4 algorithm to predict the real-time position of the satellite, i.e., its elevation and azimuth.

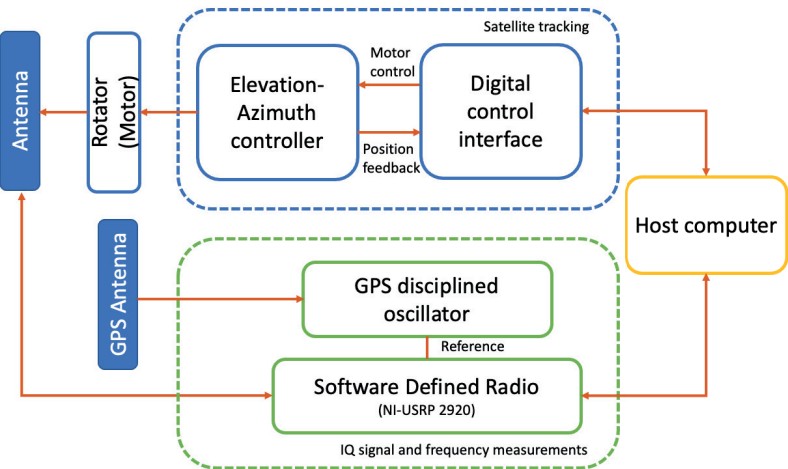

**Figure 6.** Block diagram of measurement setup for Doppler shift frequency measurements.

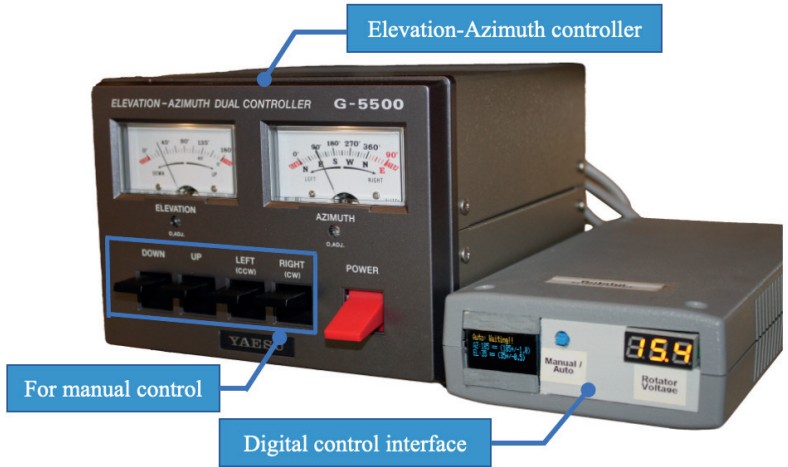

**Figure 7.** Satellite tracking antenna with digital control interface and elevation–azimuth controller connected to the antenna's rotator.

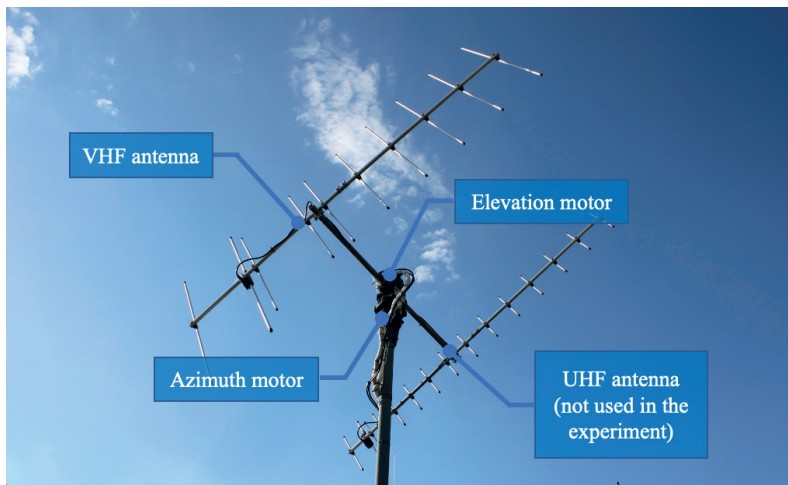

**Figure 8.** The utilized VHF antenna and the corresponding motors to control the antenna's tilt angle in the elevation and azimuth planes.

A spectrogram of the measurements is shown in Figure 9. A software-defined radio (SDR) was used to capture the APT signal from the NOAA satellite. We also attached a GPS disciplined oscillator to the SDR, which provides a very-high-accuracy reference signal as shown in Figure 10. The in-phase and quadrature (IQ) signals were collected, and a low-pass filter was applied to the signal. Then, we applied a discrete short-time Fourier transform (STFT) to the signal, $s[n]$ as follows,

$$S_m[f] = \sum_{n=-\infty}^{\infty} s[n]g[n - mR]e^{-j2\pi ft} \tag{15}$$

where $g(n)$ is the window function of $M$ length, $S_m[f]$ is the discrete Fourier transform (DFT) of the windowed signal centered at time $mR$, and $R$ is the stride size between the successive DFTs.

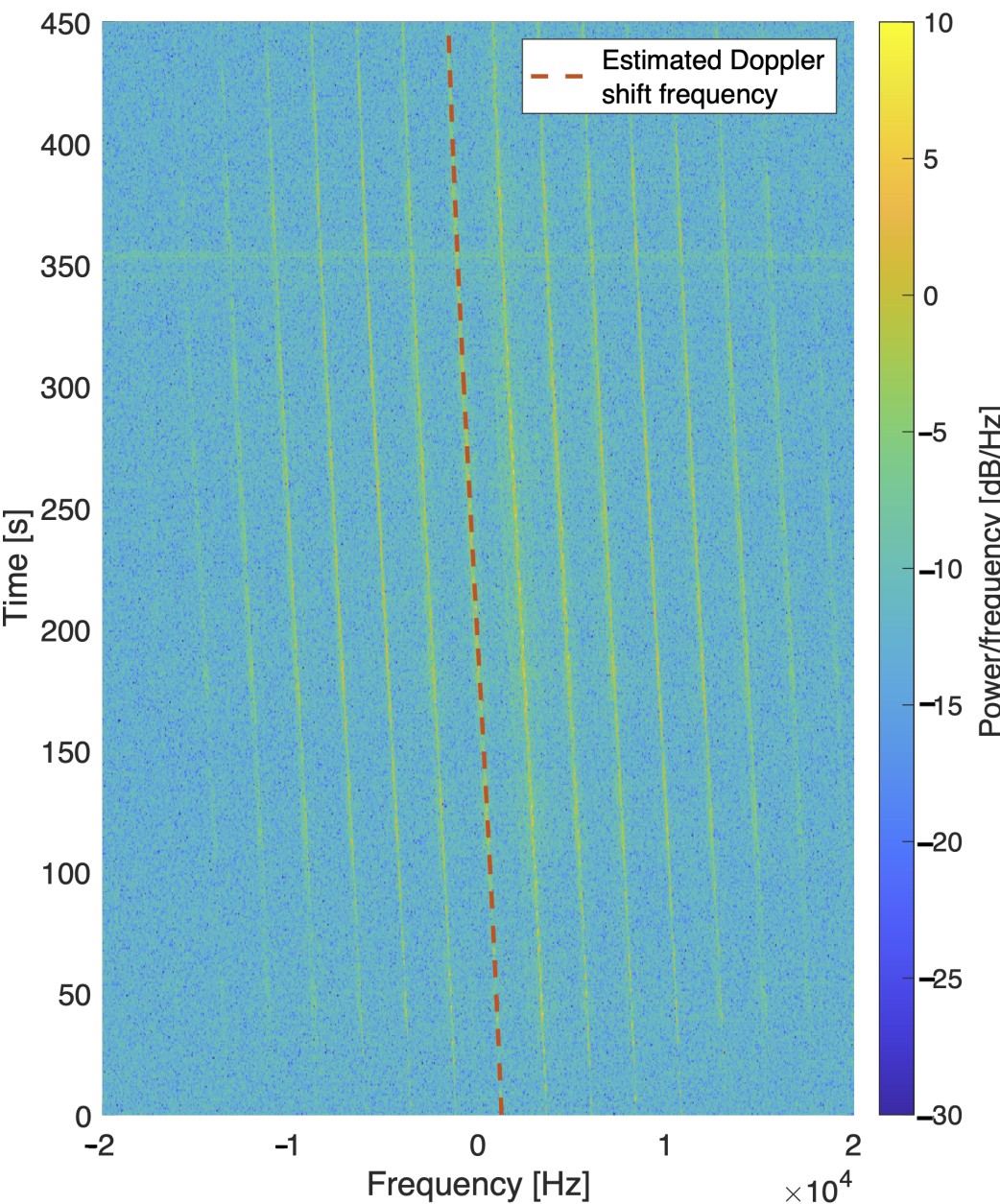

**Figure 9.** Spectrogram of the actual measurements from the NOAA-15 satellite. The orange dotted line is the estimated Doppler shift frequency.

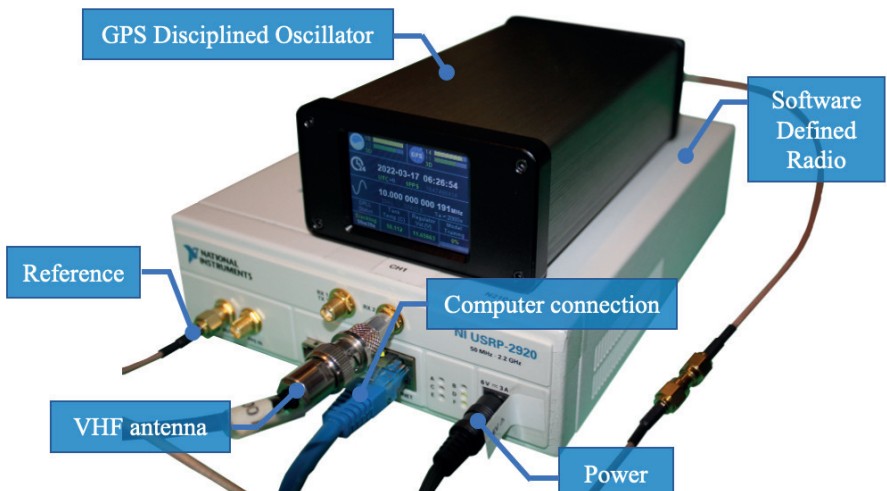

**Figure 10.** The connection between the Software-Defined Radio (National Instruments (NI) Universal Software Radio Peripheral (USRP) 2950) and the GPS-disciplined oscillator.

From the STFT of the signal, we could then search for the peak frequency to obtain the Doppler measurements. Next, to compute the measurement offset, we first needed to truncate the Doppler measurement vector and differentiate the truncated vector with respect to time. Then, we smoothed the data by fitting the data to a spline. The frequency offset was computed by taking the average of a few measurements centered at the inflection point.

Finally, we computed the empirical PDF of the Doppler measurement error. From the plot in Figure 11, it is shown that the Doppler error can be characterized as a zero-mean Gaussian distribution with a certain standard deviation, $\sigma_d$. The error distribution is Gaussian when the GPS-disciplined oscillator is utilized, whereas the error distribution is non-Gaussian for free running oscillator.

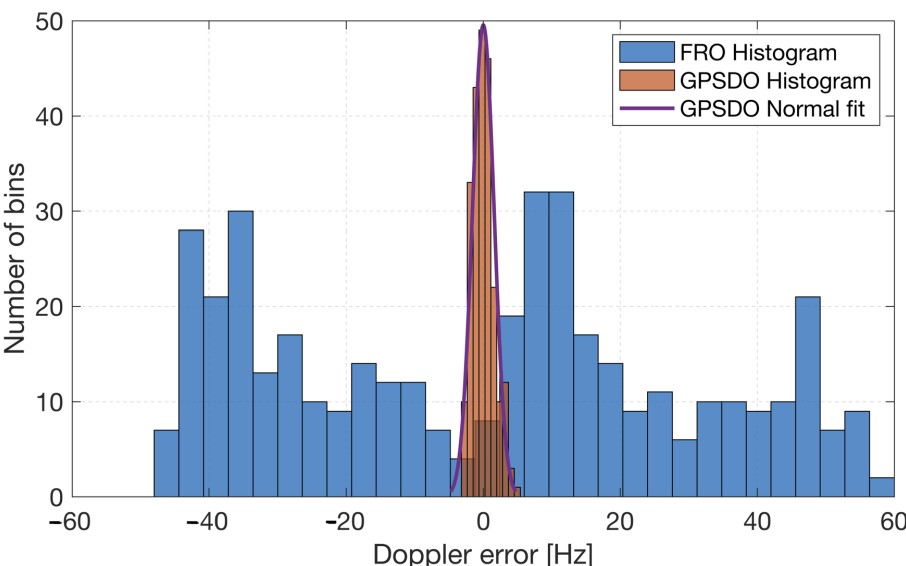

**Figure 11.** Comparison of the measured Doppler error distribution using free running (FRO) and GPS-disciplined (GPSDO) oscillators, respectively.

To measure the normality of the Doppler measurements distribution, we utilized a Kolmogorov–Smirnov test at the typical 5% significance level. From the test results, we observe that the null hypothesis was accepted, which means that the data could be considered as drawn from the normal distribution with confidence of 95%. We performed the normality test on the data collected over several passes (spanning over a few days) for the NOAA-15 satellite. Figure 12 shows a few examples of the Doppler measurement

error distribution and its corresponding *p*-value. All the five measurements passed the Kolmogorov–Smirnov test at a *p*-value of 5%.

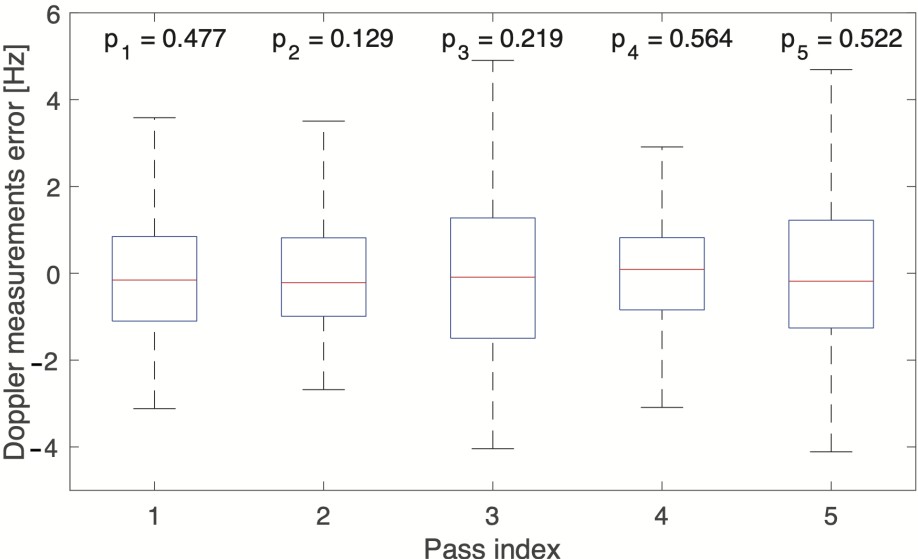

**Figure 12.** Doppler measurement error distribution for five different satellite pass examples collected over a few days. All measurements passed the Kolmogorov–Smirnov test at the 5% significance level with the corresponding *p*-values. Note that these results are for the case when the SDR is phased locked with a GPS-disciplined oscillator (GPSDO).

For a qualitative illustration, we depict an example in Figure 13 showing that the empirical cumulative distribution function (CDF) is similar to the theoretical normal CDF.

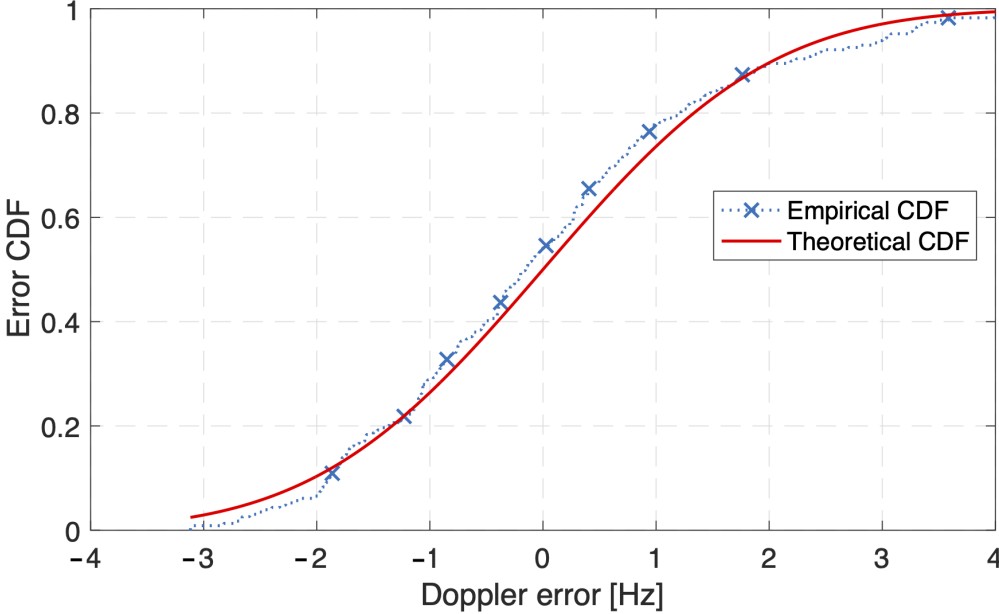

**Figure 13.** An example of empirical and theoretical Doppler error cumulative distribution functions (CDFs) for pass index 1.

In order to better understand the sources of Doppler error, we depict in Figure 14 an abstracted block diagram of the Doppler measurement chain and the corresponding sources of error at each of the steps: (i) the local oscillator of the transmitter would have an inherent phase error around a slowly drifting carrier affected by factors including the supply voltage and the ambient temperature. However for the purpose of the short measurement window,

the center of the drift, $\mu_{\text{Tx}}$ is taken as a constant, while the faster variations are captured as a normal distribution $\mathcal{N}\left(\mu_{\text{Tx}}, \sigma_{\text{Tx}}^2\right)$ [32]. (ii) The signal at the receiver is shifted by the true Doppler value of $f_d$, then (iii) it is mixed with the receiver's local oscillator that also exhibits a similar behavior to the transmitter's local oscillator with a normal model of $\mathcal{N}\left(\mu_{\text{Rx}}, \sigma_{\text{Rx}}^2\right)$. Accordingly, the final Doppler error can be expressed as follows,

$$\epsilon_1 = \epsilon_{\text{Tx}} + \epsilon_{\text{Rx}} + \epsilon_{\text{E}}, \tag{16}$$

where $\epsilon_{\text{Tx}} \sim \mathcal{N}\left(\mu_{\text{Tx}}, \sigma_{\text{Tx}}^2\right)$ and $\epsilon_{\text{Rx}} \sim \mathcal{N}\left(\mu_{\text{Rx}}, \sigma_{\text{Rx}}^2\right)$ are the error caused by the Tx and Rx local oscillators, and $\epsilon_{\text{E}}$ is the error caused by the estimation algorithm. Since the signal should incur no Doppler shift at the highest elevation point (*inflection point*), we could estimate the constant drifts and subtract it from the Doppler measurement vector. In this case, the Doppler error reduces to,

$$
\begin{aligned}
\epsilon_2 &= \mathcal{N}\left(0, \sigma_{\text{Tx}}^2\right) + \mathcal{N}\left(0, \sigma_{\text{Rx}}^2\right) + \mathcal{N}\left(0, \sigma_{\text{E}}^2\right), \\
&= \mathcal{N}\left(0, \sigma_{\text{Tx}}^2 + \sigma_{\text{Rx}}^2 + \sigma_{\text{E}}^2\right).
\end{aligned}
\tag{17}
$$

Furthermore, the error at the receiver is practically negligible because of the use of the GPS disciplined oscillator which is in the range of $\mu$Hz. Accordingly, a simpler error model can be expressed as $\epsilon_d \sim \mathcal{N}\left(0, \sigma_{\text{Tx}}^2 + \sigma_{\text{E}}^2\right)$, where by going back to (5) we note that $\sigma_d^2 = \sigma_{\text{Tx}}^2 + \sigma_{\text{E}}^2$. In the IoT localization context, the transmitter would be the ground IoT device, while the receiver is the satellite. Hence, it is reasonable to assume/stipulate that the satellite has access to a highly stable oscillator to minimize the localization error.

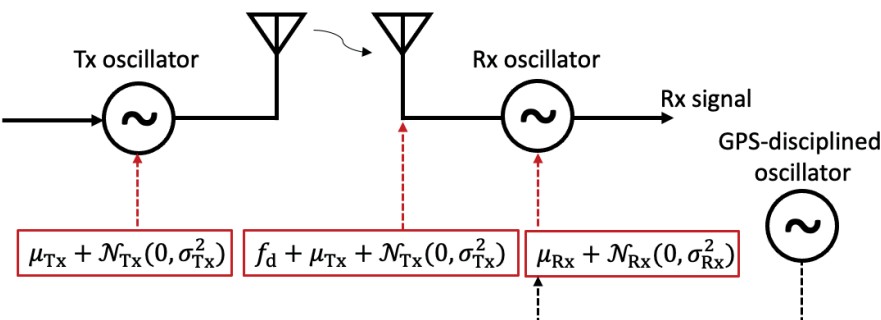

**Figure 14.** Block diagram of Doppler measurements and the sources of error.

### 6.2. Localization Simulation and Performance

In the simulation, we studied low-Earth orbit satellites in a Walker-star constellation at an altitude of 833 km. The constellation was modeled using the *Satellite Communications toolbox* by MATLAB, with orbits created as indicated in Section 3.1. The simulated period is one complete orbit. We extracted the true AoA from the geometric model, introduced a random error to represent the AoA measurements, and computed the true Doppler measurements by differentiating the geometric slant range and introduced a random error accordingly. Furthermore, Figure 15 shows an example snapshot of the constellation as seen by a ground IoT device; the satellites that are in use for localization is labeled as a green dot, where the elevation angles threshold of 15° is used. This threshold is set to ensure the emulation of a practical scenario, where near-horizon signals would typically incur much higher atmospheric absorption and shadowing [39]. Furthermore, we demonstrate the beginning and ending points of the contributing satellite in the localization as depicted in Figure 16.

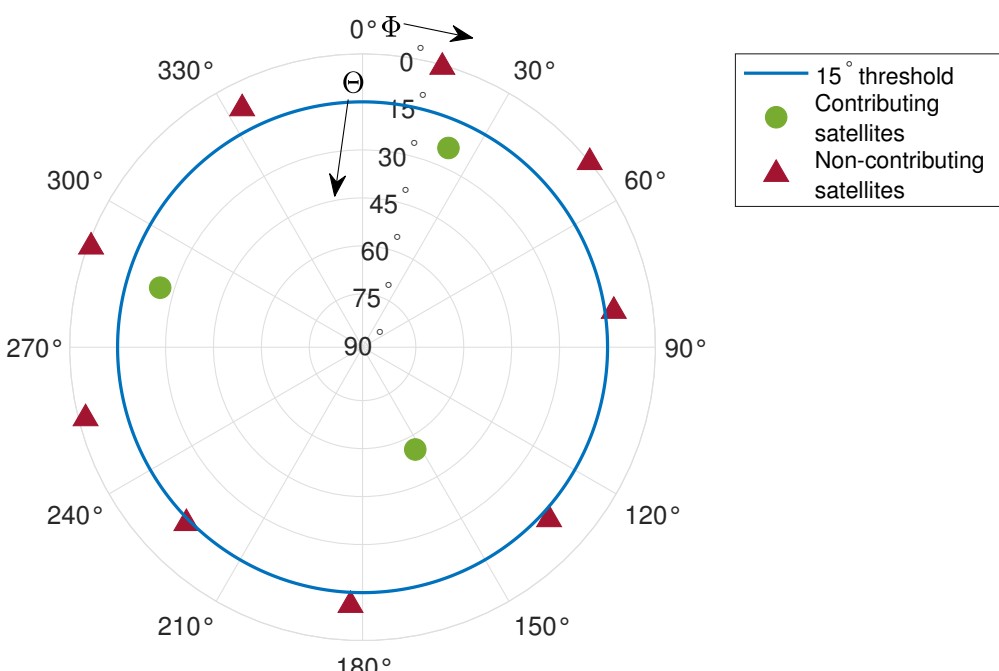

**Figure 15.** An example snapshot of the contributing satellites that are above the 15° elevation threshold with respect to a ground IoT device. The collected measurements from these satellites are used in the localization. Non-contributing satellites that are below the threshold are marked by red triangles.

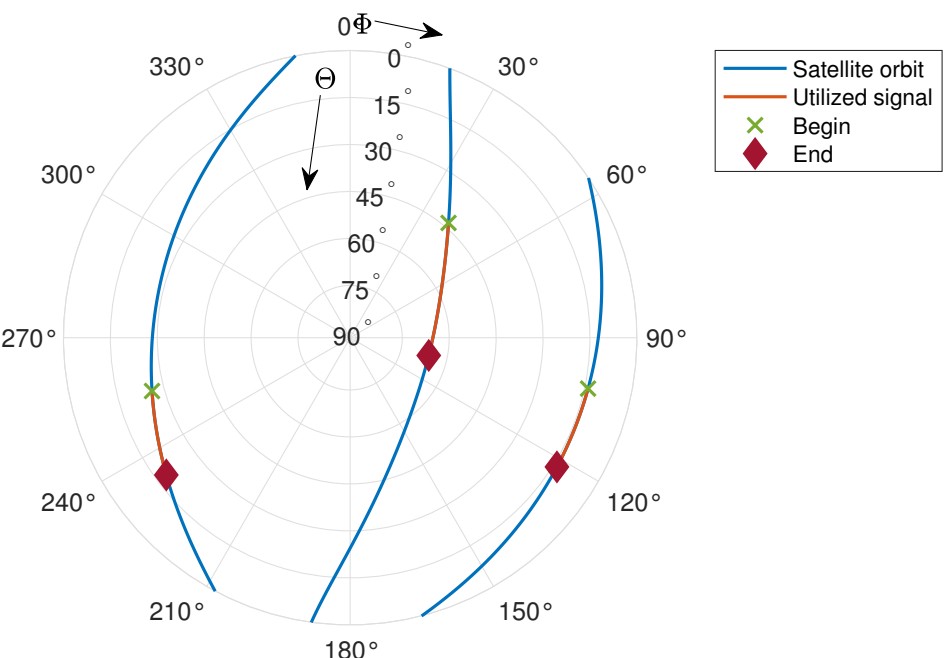

**Figure 16.** Satellite orbits that are above the elevation threshold and the corresponding location at the beginning and end of the utilized segment for signal measurements in the localization algorithm.

To visualize that the derived joint likelihood function has a maximum value at the actual position of the ground IoT device, we plot the log likelihood function as shown in Figure 17. This is achieved by computing the likelihood for all of the points ranging from the latitude of $-39.8°$ to $-25°$ and longitude of $143.5°$ to $149.5°$. From the plot, it is evident that the derived likelihood function has a high likelihood at the ground truth. It is important to note that in the actual localization process presented in the next subsection,

we used the stochastic optimizer rather than calculating the likelihood by brute force, as the latter is much more computationally costly.

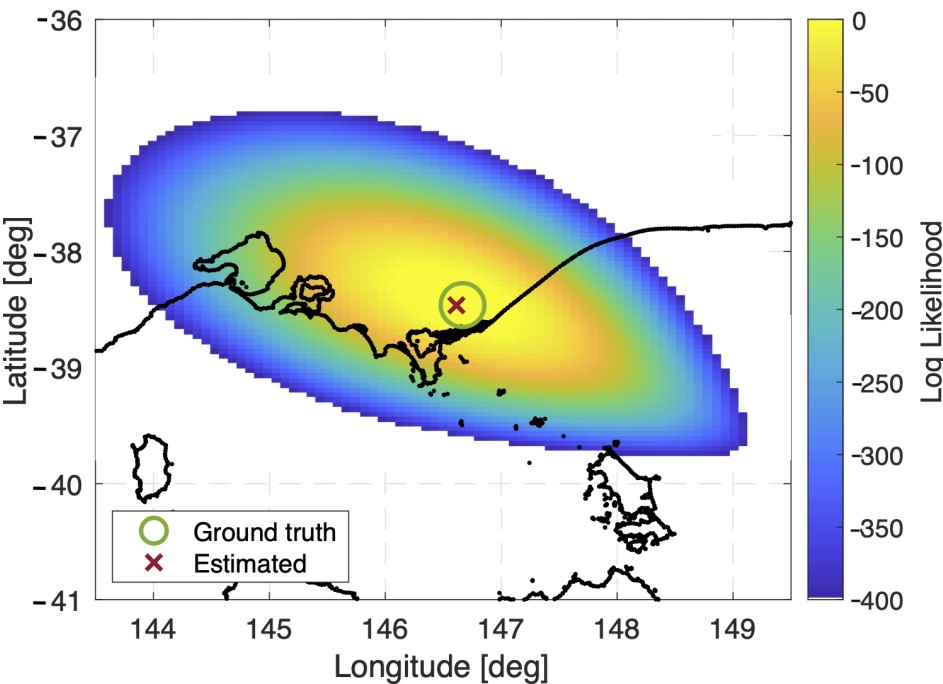

**Figure 17.** An example of the joint AoA and Doppler log likelihood function, showing the ground truth point.

We applied a stochastic optimizer, namely simulated annealing (SA), to search for a state that efficiently maximizes the likelihood function. SA was chosen due to its performance compared to the other optimizers such as the genetic algorithm (GA), surrogate optimizer (SO), and particle swarm (PSW). The SA optimizer is developed from the physical process of changing material structure by firstly heating the material to its annealing temperature then allowing it to cool gradually. The physical process attempts to achieve the minimum possible internal energy naturally. At high temperatures, the algorithm explores the solution space extensively, thus allowing for uphill moves of the cost function gradient. As the temperature decreases, the algorithm becomes more selective. It starts favoring downhill moves, but still allows for occasional uphill jumps with a certain probability [40]. The SA optimizer follows this physical process to optimize the parameters in a model, which in our case, minimizes the cost function.

In contrast, the PSW optimizer uses a swarm of particles to explore the solution space where each particle represents a potential solution. Each particle adjusts its position based on its own experience and the collective knowledge of the swarm. However, it is crucial to tune the parameters such as inertia weight, cognitive and social coefficients, and swarm size to obtain an effective PSW optimizer [41]. Due to this, PSW may face challenges like premature convergence where the algorithm is stuck in a local optima.

Furthermore, the GA optimizer is based on the individual's fitness of a population, which is evaluated depending on how well it solves the optimization problem. The fitter individuals have a higher chance of being selected, and a pair of individuals will create new offspring to introduce diversity, thus helping to explore the solution space. Furthermore, random changes are applied to some individuals to introduce additional mutations. The process is repeated for several generations. Similar to the PSW optimizer, it is vital to find the right balance of the parameters that affect the convergence [42]. The GA optimizer is susceptible to premature convergence, where the population stops evolving, thus causing stagnation. Accordingly, SA optimizers are preferred for a global search over PSW and GA optimizers.

In addition, the SO optimization process starts with the creation of a surrogate model, often based on the initial evaluations of the objective function. A set of initial samples is selected and evaluated using the surrogate model. The model gets refined as more points are sampled and evaluated. The same model is then utilized to propose new candidate solutions. These solutions are selected based on the predictions made by the surrogate model and evaluated using the true objective function. The proposal and evaluation of candidate solutions is repeated iteratively to converge to the best solution [43]. For the SO optimizer, the quality of surrogate model plays a critical role in the convergence. The success of the method depends heavily on the effectiveness of the surrogate model. Since it requires a well-built model, the SO optimizer is more problem-specific, whereas the SA optimizer is versatile and able to handle various types of optimization.

Moreover, deterministic optimization methods involve calculating derivatives of the objective function. The derivatives provide information about the direction and rate of change of the function at a given point. Based on the information, a search direction is determined. The current solution is updated along the chosen search direction [44]. However, deterministic methods struggle in dealing with non-smooth, stochastic, or non-convex objective functions; thus, they can get stuck in local minima. The SA optimizer performs better than other deterministic methods for this stochastic optimization problem due to its ability to avoid the local minimum by allowing a move to a higher cost function with some probability [45].

To cater to the IoT device limitation of short and intermittent transmission, the simulation was run only using a short measurement time ($\sim$75 s) from the available satellites. Firstly, we illustrate the localization performance using a stochastic optimizer (simulated annealing) with an increasing number of satellites involved in the localization as depicted in Figure 18. The figure shows that as the number of contributing satellites increases, the localization error decreases. This figure also shows the impact of Doppler error on the localization error where, as expected, a higher Doppler error leads to a lower localization accuracy.

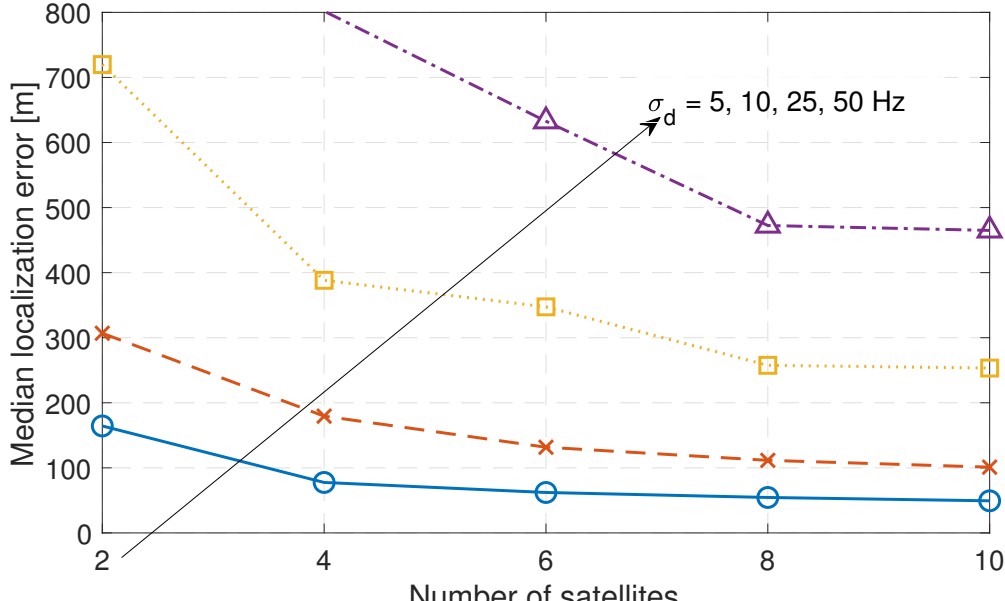

**Figure 18.** The median localization error with increasing satellites for the stochastic optimizer. Different curves represent the Doppler standard deviation $\sigma_\mathrm{d} = 5, 10, 25,$ and 50 Hz and $\sigma_\Phi, \sigma_\Theta = 1°$; the results are averaged over 300 runs.

To understand the impact of the number of measurement samples taken over separate time instances on the localization error, we depict an example in Figure 19, where the measurements are separated by 5 s. As expected, as more samples are used in the algorithm, the localization error is lower. With a combination of four or more satellites and 15 or more

measurement sample points, the achievable median localization error is less than 150 m. Note that for non-stationary IoT devices, taking measurements over long periods of time might result in *target migration* and thus reduce the localization accuracy.

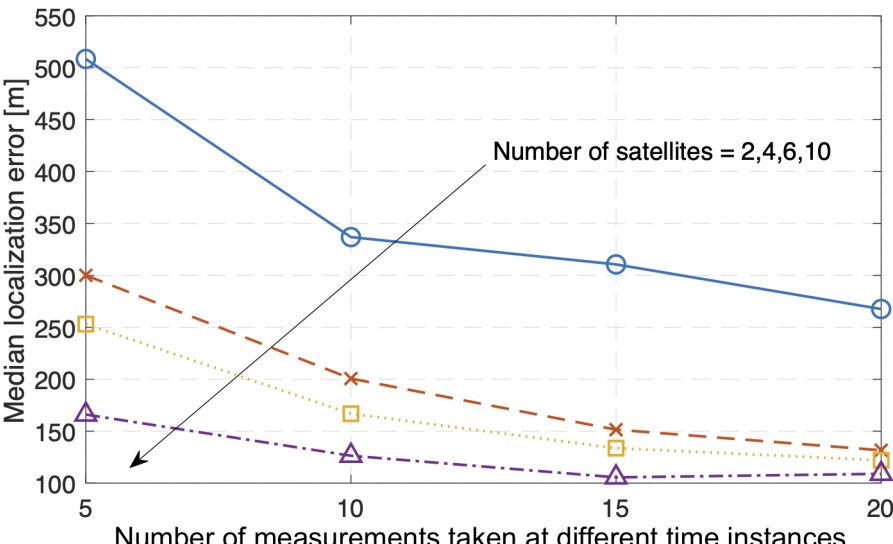

**Figure 19.** The median localization error for a varying number of measurements taken at intervals separated by 5 s and different curves represent the number of satellites with Doppler standard deviation, $\sigma_d = 10$ Hz and AoA standard deviation $\sigma_\Phi$, $\sigma_\Theta = 1°$; the results are for 300 runs.

Furthermore, to observe the effect of the AoA estimation error on the localization performance, we increased the AoA standard deviation to a range of 0.01 to 1° as indicated in Figure 20. The figure shows that as the AoA error increases, the localization accuracy reduces. From the plot, it can also be observed that the effect of AoA standard deviation of more than 0.05° is more prominent when the Doppler standard deviation is more than 15 Hz. In contrast, when the AoA standard deviation is very small, such as 0.01°, the Doppler standard deviation has little effect on the localization performance. From the results, it is evident that at lower errors in AoA measurements, the localization performance improves significantly even when the Doppler measurements have large errors.

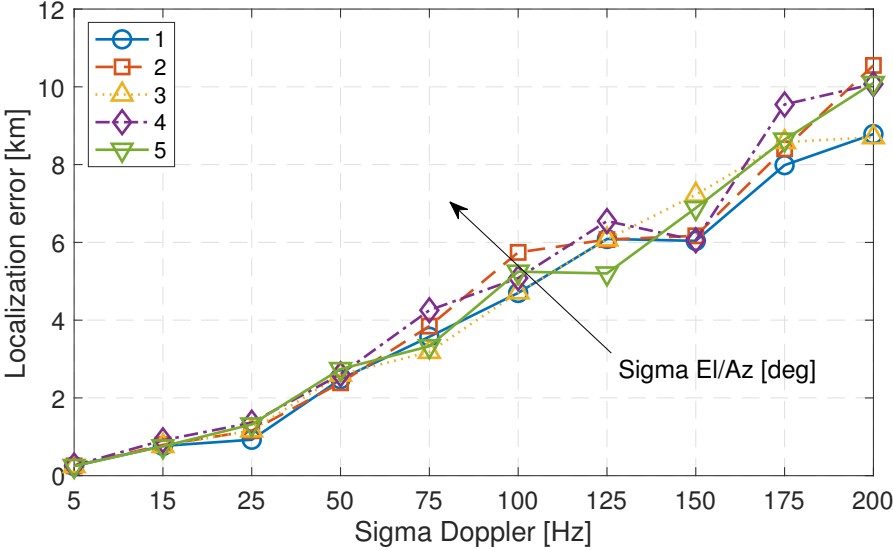

**Figure 20.** The median localization error for varying Doppler standard deviation, $\sigma_d$ and azimuth, $\sigma_\Phi$ and elevation, and $\sigma_\Theta$ AoA deviation using six satellites and 75 s of measurements; the results are deduced from 300 runs.

To observe the merit of the joint utilization of Doppler and AoA, we depict in Figure 21 three cases: (i) when only using Doppler measurements, (ii) when only using AoA measurements, and (iii) when using both Doppler and AoA jointly for localization. Given a Doppler standard deviation, $\sigma_d$ at 5 Hz and AoA deviation, $\sigma_\Phi, \sigma_\Theta$ at 0.01°, the median localization error for Doppler is 64.29 m, AoA is 43.78 m, and the joint measurement is 34.55 m. From the results, it can be noted that combining the AoA measurements with the Doppler measurements has a positive effect on the localization performance. It not only reduces the median localization error but also narrows the error distribution. Therefore, it is clearly beneficial to utilize both the Doppler and AoA measurements for ground IoT device localization using a constellation of satellites.

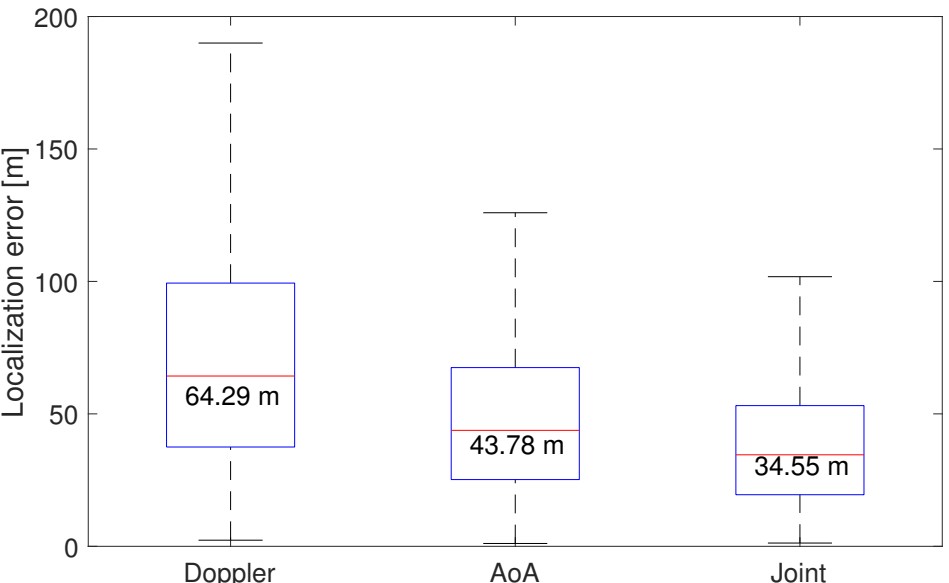

**Figure 21.** The localization error using Doppler, AoA and joint Doppler-AoA measurements for Doppler standard deviation, $\sigma_d = 5$ Hz and azimuth, $\sigma_\Phi = 0.01°$ and elevation, and $\sigma_\Theta = 0.01°$ deviation using six satellites and 75 s of measurements; the results are based on 1600 runs.

## 7. Conclusions

This paper proposes a IoT-over-satellite-based localization framework using both AoA and Doppler shift frequency patterns. A likelihood function was derived from the measurements based on the Gaussian approximation of the Kent distribution for AoA and the Gaussian distribution for Doppler. To estimate the location of a ground IoT device, a stochastic optimizer was utilized to maximize the likelihood function, or, in other words, to minimize the negative log likelihood function. An experimental satellite ground station was utilized to obtain realistic Doppler measurements of NOAA LEO satellites in order to support the Gaussian model of Doppler error. From the simulation results, it is observed that the achievable median localization error is less than 50 m when using the best configuration (highest number of satellites and measurement points and lowest number of measurement deviations). Furthermore, it is shown that the combination of AoA and Doppler measurements improves the localization performance significantly. In the future work, we suggest the satellite's attitude errors to be included in the localization estimation to accommodate for a wider range of scenarios.

**Author Contributions:** Conceptualization, I.S.M.H. and A.A.-H.; Methodology, I.S.M.H. and A.A.-H.; Software, I.S.M.H.; Formal analysis, I.S.M.H.; Investigation, I.S.M.H. and A.A.-H.; Writing—original draft, I.S.M.H.; Writing—review & editing, A.A.-H.; Visualization, I.S.M.H.; Supervision, A.A.-H. All authors have read and agreed to the published version of the manuscript.

**Funding:** This research was funded by the Australian Government Research Training Program.

**Data Availability Statement:** The data presented in this study are available on request from the corresponding author. The data are not publicly available due to a very large simulation data size.

**Conflicts of Interest:** The authors declare no conflict of interest.

## Abbreviations

| Symbol | Definition | Value (Unit) |
|---|---|---|
| $N$ | Number of satellites | 288 |
| $P$ | Number of orbital planes | 12 |
| $F$ | Phasing parameter | - |
| $j$ | Orbital plane index | - |
| $l$ | Order within orbital plane | - |
| $\Omega$ | Right ascension of ascending node | $0$–$2\pi$ |
| $\nu$ | Initial true anomaly | $0$–$2\pi$ |
| $S$ | Number of satellites on an orbital plane | 24 |
| $v_r$ | True radial velocity | variable (m/s) |
| $\rho$ | Slant distance between a satellite and a ground IoT device | variable (m) |
| $t$ | Time variable | - |
| $\Delta t$ | Simulation time step | 5 (s) |
| $f_\mathrm{d}, f_r$ | Classical & relativistic Doppler shift frequency | - (Hz) |
| $c$ | Speed of light | 299,792,458 (m/s) |
| $f$ | Center operating frequency | 2 (GHz) |
| $\mu_\mathrm{d}$ | True Doppler shift frequency | - (Hz) |
| $\sigma_\mathrm{d}$ | Standard deviation of Doppler error | - (Hz) |
| $\Phi$ | Azimuth angle of arrival | - (°) |
| $\Theta$ | Off-nadir angle of arrival | - (°) |
| $\lambda_S$ | Latitude of the source (ground IoT device) | - (°) |
| $\phi_S$ | Longitude of the source (ground IoT device) | - (°) |
| $\mathbf{x_a}$ | Position vector | - |
| $\mathbf{M}$ | Coordinate transformation matrix | refer to (6) |
| $\mathbf{x_S}$ | Satellite coordinate in ECEF frame | - |
| $\mathbf{x_G}$ | Ground IoT device coordinate in ECEF frame | - |
| $\kappa$ | Concentration parameter in Kent distribution | - |
| $\beta$ | Ovalness parameter in Kent distribution | - |
| $I_3$ | Identity matrix of size 3 | - |
| $\sigma_\Phi$ | Standard deviation of azimuth AoA error | - (°) |
| $\sigma_\Theta$ | Standard deviation of off-nadir AoA error | - (°) |
| $\mathbf{x}$ | State vector (latitude and longitude of ground IoT device) | - (°) |
| $\mathbf{Z}^{(k)}$ | Doppler and AoA measurement vector | - |
| $k$ | Discrete-measurement index | - |
| $g_1(.)$ | Doppler likelihood function | - |
| $g_2(.)$ | AoA likelihood function | - |
| $\mu_\Phi$ | True azimuth angle of arrival | - (°) |
| $\mu_\Theta$ | True off-nadir angle of arrival | - (°) |
| $p(\mathbf{Z}^{(k)}|\mathbf{x})$ | Joint likelihood of Doppler and AoA | - |

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
