# Peer review of "Satellite-Based Localization of IoT Devices Using Joint Doppler and Angle-of-Arrival Estimation"

_remotesensing, doi:10.3390/rs15235603_

Round 1

Reviewer 1 Report

Comments and Suggestions for Authors

In this letter, the author proposes a IoT-over-satellite-based localization framework using joint Doppler shift frequency and AoA Estimation. The simulation experiment results shown that the combination of AoA and Doppler measurements improves the localization performance significantly compare with only using Doppler measurements and only using AoA measurements patterns. The idea of this research is impressive. Organization of the manuscript is correct and the language is understandable, but the presentation has some drawbacks. The highlights are clear and informative. The abstract contains all necessary information. The introduction provides adequate background of the topic. Proposed method is presented in detail. Performed experiments are properly described. I think the manuscript can be accepted for publication after a minor revision.

I have some remarks:

1. In the process of solving the maximum likelihood function, the stochastic optimizer chooses simulated annealing, but the introduction of its core process is less. If possible, it is suggested that the author simultaneously compare the deterministic methods, particle swarm optimization, genetic algorithm, differential evolution and other algorithms, and show the convergence process of its cost function values, so as to highlight the advantages of simulated annealing.

2. The legend of different curves in Figures 18 and 19 is missing, and it is not easy for the reader to distinguish them.

3. There are errors in the reference information, please check it carefully.

Author Response

Please see the attachment. Thank you. Note that the citation style in the response document is not the same as in the manuscript. In the manuscript, the citation is in numbers whereas in the response document is in Author's name, year format. 

Reviewer 2 Report

Comments and Suggestions for Authors

This paper presented a IoT-over-satellite-based localization framework using both AoA and Doppler shift frequency measurements. The method is applicable, but the statement of simulation is not so clear to me.

1. In line 5, "receiving satellite" seems a bit wired to me, I believe you were saying "received satellite signal".

2. In Fig. 3, the unit of time is missing. Please fix it.

3. If I understand correctly, the Doppler shift measurement acts as a function of position following both equation (3) and equation (4),  however, I didn't find how the value of \delta_t is specified in the simulation. Please include it.

4. My confusion mainly focused on Fig.19. which is used to show the more measurement the higher the accuracy. I don't quite get what you mean by the number of satellites and the number of measurements. As far as I understand, the number of satellites is the number of measurements, since every time you receive a satellite signal, you can obtain the satellite position for AoA and Doppler shift measurements. Maybe I missed something here, could you make it more clear?

5. The simulation demonstrated the authors' point , but a experiment would be more convincing. I'm wondering if there's any chance to include an experiment for the localization accuracy test of the algorithm? If it's unnecessary, please explain why.

Reviewer 3 Report

Comments and Suggestions for Authors

REVISION MANUSCRIPT Remote Sensing- 2696737: Satellite-based Localization of IoT Devices using Joint Doppler and Angle-of-Arrival Estimation.

General comments:

The authors argued that GNSS may not be the best choice for some Internet-of-Things (IoT) applications due to the incurred power consumption and cost. Hence, the authors presented an alternative satellite-based localization method exploiting the signature of Doppler shifts and angle of arrival measurements as seen by a receiving satellite. To accomplish that, the authors proposed utilizing a stochastic optimizer to search for the global minimum of the MSE which represents the location of the ground IoT device as estimated by the satellite platform. Therefore, I found the research interesting, and I believe some minor suggestions and revisions need to be applied in order to consider it for publication.

Specific comments:

First, IoT devices and how they achieve localization through Doppler measurements and the method of estimating the angle of arrival are a bit unknown, in general. Considering the final positioning as the objective of this work, I think that currently, most IoT devices have a GNSS chip to position themselves and the authors' justification, where they say "that positioning with GNSS is expensive and consumes a lot of energy" is somewhat vague, devices, for example cell phones have an energy saving system when GNSS is used, And the prices are relatively low. So, the authors' justification is very poor, there must be other reasons why they do that work. In addition, the accuracy of 50 meters that they report in their conclusions is very high, with any GNSS navigator you can get better accuracy.

Round 2

Reviewer 2 Report

Comments and Suggestions for Authors

The authors have addressed almost all my issues.

The only thing is that, in the the response letter, the authors mentioned that "the number of measurements are the number of measurements points for each satellite in Figure 19". It's still not so clear to me. I assume the different measurements points means the measurements over different time instant? Please illustrate it more clearly, I'm worried if readers will get the same confusion as me.

In my view, the paper is ready to go except this small confusion.

Author Response

Thank you for the encouraging feedback. Please see the attachment for the response. 
